# The Role of FAS Receptor Methylation in Osteosarcoma Metastasis

**DOI:** 10.3390/ijms241512155

**Published:** 2023-07-29

**Authors:** Jiayi M. Sun, Wing-Yuk Chow, Gufeng Xu, M. John Hicks, Manjula Nakka, Jianhe Shen, Patrick Kwok Shing Ng, Aaron M. Taylor, Alexander Yu, Jason E. Farrar, Donald A. Barkauskas, Richard Gorlick, Jaime M. Guidry Auvil, Daniela Gerhard, Paul Meltzer, Rudy Guerra, Tsz-Kwong Man, Ching C. Lau

**Affiliations:** 1Program of Quantitative and Computational Biosciences, Baylor College of Medicine, Houston, TX 77030, USA; monikasun88@gmail.com (J.M.S.); aaron.taylor@jax.org (A.M.T.); tman@bcm.edu (T.-K.M.); 2Department of Pediatrics-Oncology, Baylor College of Medicine, Houston, TX 77030, USA; wxchow@txch.org (W.-Y.C.); gufeng_xu@163.com (G.X.); mxnakka@txch.org (M.N.); jjshen77024@gmail.com (J.S.); alexanderyu@me.com (A.Y.); 3Texas Children’s Cancer and Hematology Center, Houston, TX 77030, USA; 4Department of Pathology, Texas Children’s Hospital, Baylor College of Medicine, Houston, TX 77030, USA; hicks@bcm.edu; 5The Jackson Laboratory for Genomic Medicine, Farmington, CT 06032, USA; patrick.ng@jax.org; 6Arkansas Children’s Research Institute and Department of Pediatrics, University of Arkansas for Medical Sciences, Little Rock, AR 72205, USA; jefarrar@uams.edu; 7Department of Population and Public Health Sciences, Keck School of Medicine, University of Southern California, Los Angeles, CA 90033, USA; barkausk@usc.edu; 8Division of Pediatrics, The University of Texas MD Anderson Cancer Center, Houston, TX 77030, USA; rgorlick@mdanderson.org; 9Office of Cancer Genomics, National Cancer Institute, Bethesda, MD 20892, USA; jaime.guidryauvil@nih.gov (J.M.G.A.);; 10Genetics Branch, Center for Cancer Research, National Cancer Institute, National Institutes of Health, Bethesda, MD 20892, USA; pmeltzer@mail.nih.gov; 11Department of Statistics, Rice University, Houston, TX 77005, USA; rguerra@rice.edu; 12Center for Cancer and Blood Disorders, Connecticut Children’s Medical Center, Hartford, CT 06106, USA

**Keywords:** osteosarcoma, FAS receptor, DNA methylation, 5-azacytidine, prognosis

## Abstract

Osteosarcoma is the most frequent primary malignant bone tumor with an annual incidence of about 400 cases in the United States. Osteosarcoma primarily metastasizes to the lungs, where FAS ligand (FASL) is constitutively expressed. The interaction of FASL and its cell surface receptor, FAS, triggers apoptosis in normal cells; however, this function is altered in cancer cells. DNA methylation has previously been explored as a mechanism for altering FAS expression, but no variability was identified in the CpG island (CGI) overlapping the promoter. Analysis of an expanded region, including CGI shores and shelves, revealed high variability in the methylation of certain CpG sites that correlated significantly with FAS mRNA expression in a negative manner. Bisulfite sequencing revealed additional CpG sites, which were highly methylated in the metastatic LM7 cell line but unmethylated in its parental non-metastatic SaOS-2 cell line. Treatment with the demethylating agent, 5-azacytidine, resulted in a loss of methylation in CpG sites located within the FAS promoter and restored FAS protein expression in LM7 cells, resulting in reduced migration. Orthotopic implantation of 5-azacytidine treated LM7 cells into severe combined immunodeficient mice led to decreased lung metastases. These results suggest that DNA methylation of CGI shore sites may regulate FAS expression and constitute a potential target for osteosarcoma therapy, utilizing demethylating agents currently approved for the treatment of other cancers.

## 1. Introduction

Osteosarcoma (OS) is a type of primary malignant bone tumor that is most commonly seen in children and adolescents. While OS is a relatively rare tumor, with an approximate incidence of five in one million per year for those under the age of 20, it is the most common form of bone cancer, accounting for nearly 58% of all cases in that age range [1]. Standard therapy for OS patients includes neoadjuvant chemotherapy and surgical resection followed by post-operative chemotherapy [2]. At the time of definitive surgery, histologic response to neoadjuvant chemotherapy is used as a prognostic indicator for overall survival [3,4]. While OS patients who respond well to standard treatment have a favorable 6-year survival rate of 70%, high-risk patients who have metastatic disease or respond poorly to the neoadjuvant regimen have a much poorer survival rate of 30% [5,6,7,8]. In the past few decades, there has been very little improvement in the long-term survival of osteosarcoma patients, largely due to the lack of prognostic biomarkers to predict patients with high-risk disease and no alternative or targeted treatments for these patients [6]. Modification of the post-operative chemotherapy regimen, including the use of novel chemotherapeutic agents, has not led to measurable improvements in survival [9,10].

DNA methylation is a form of epigenetic regulation where methyl groups are added to the cytosine of cytosine-guanine dinucleotides (CpG sites). Methylation of CGIs plays an important role in the regulation of downstream gene transcription through altering the accessibility of transcription factors and other components of gene regulation [11,12]. Approximately 40% of known genes contain at least one CGI in the promoter region [13]. Higher DNA methylation in CGIs is traditionally associated with gene silencing. More recently, evidence has emerged indicating a role for DNA methylation in regions outside the CpG island body, such as the shores (~2 kb away from the CGI) and the gene body, in addition to a role in controlling downstream gene expression as well. Methylation at CpG island shores is associated with cancer-specific alterations in colon cancer and has been shown to negatively regulate gene expression in breast cancer [14,15]. Gene body methylation is typically correlated positively with gene expression and may be associated with nucleosome destabilization, resulting in decreased transcriptional efficiency [16,17]. Abnormalities in DNA methylation characterize cancer subgroups and alter the expression of tumor suppressors or oncogenes, thus playing an important role in tumor pathogenesis and progression [18,19,20]. As more is known about the DNA methylome of specific cancers and the role of DNA methylation in transcriptional activation or repression in general, the potential for using agents targeting aberrant methylation as alternative cancer therapies increases.

FAS (CD95/APO-1) is a member of the tumor necrosis factor family of receptors. Interaction between FAS and its ligand (FASL) triggers downstream apoptosis through the activation of caspase 8 and formation of the death-inducing signaling complex [21]. Silencing of FAS expression results in decreased interaction with FASL, leading to reduced apoptosis, and aberrant DNA methylation is one possible mechanism through which this may occur. The FAS promoter contains a CGI that extends into the first intron and overlaps binding sites for both NF-kB and p53 [22,23]. Negative correlation between methylation and expression of FAS has been reported in several cancers such as colon cancer, small cell lung carcinoma and a variant of T-cell lymphoma, Sezary Syndrome [24,25,26,27]. Chemotherapy agents that modify aberrant DNA methylation such as 5-aza or 5-aza-2′deoxycitidine (decitabine) have been approved for treatment of myelodysplastic syndrome [28]. Treatment with 5-aza or other demethylating agents led to the re-expression of the FAS receptor and restoration of proapoptotic function in other cancers and may be a potential novel treatment for osteosarcoma [24,26,27,29].

FAS expression is important in the context of OS because the majority of OS metastases arise in the lungs, where FASL is constitutively expressed. LM7, a metastatic osteosarcoma cell line, has significantly reduced FAS expression in comparison to its non-metastatic parental cell line, SaOS-2 [30]. Transfection of the FAS gene into LM7 cells leads to decreased tumor formation in nude mice [31]. In an immunohistochemistry study of lung nodules specifically from metastatic osteosarcoma patients, 60% were negative for FAS [32]. Based on these data, it was hypothesized that OS cells expressing higher levels of FAS would have a smaller chance to establish metastasis in the lungs, due to a higher chance of interaction with FASL. By identifying a mechanism through which FAS expression is modified, it may be possible to use an alternative form of therapy to treat or prevent metastatic disease by inducing a higher level of FAS expression in tumor cells. In OS, DNA methylation of the CGI body was previously found to be invariant and did not associate with changes in FAS expression [33]. However, studies in other cancers profiling the shore regions of the promoter CGI have identified variant methylation at CpG sites, which were negatively correlated with FAS expression [26]. Therefore, we decided to fully utilize the expanded coverage of Illumina’s HumanMethylation450k (HM450) array to evaluate FAS promoter methylation.

In this study, we comprehensively profiled FAS promoter methylation and expression in OS patients and cell lines. Our results indicated that FAS methylation is highly variant in CpG sites located in the island shores and correlated well with FAS mRNA expression. We further validated the status of these sites in osteosarcoma cell lines using bisulfite sequencing. We carried out functional studies using 5-aza to demethylate these CpG sites, which led to increased FAS expression and decreased cell migration. Finally, using an orthotopic xenograft mouse model of OS, we observed the decreased formation of lung metastases in mice injected with 5-aza treated metastatic LM7 cells. Overall, our results indicate that FAS promoter methylation in the CpG island shore may regulate the expression of FAS, thereby altering the metastatic potential of osteosarcoma patients.

## 2. Results

### 2.1. FAS Expression in Patient Tumors Correlates with Clinical Outcome

Initially, we identified FAS as a differentially expressed gene between unsupervised clusters of the tumor mRNA expression array data. Among the OS tumor samples, we found that FAS expression was variant with a standard deviation (SD) 0.93 (log2 intensity), which puts the FAS gene approximately in the top 6% of the SD distribution of all genes on the array (Figure 1A). In a Cox proportional hazards model, low FAS mRNA expression (stratified at the 25th percentile) was significantly associated with poorer overall survival, with a 5-year survival rate of 74% versus 37% (Figure 1B). Event-free survival was similarly associated with FAS expression. Patients with low FAS expression had a 58% chance of developing metastasis at 5 years in comparison to a 36% chance among those with a high expression of FAS (Figure 1C). We also tested the observed survival difference based on FAS expression in the non-overlapping 81-sample cohort from the Strategic Partnering to Evaluate Cancer Signatures (SPECS) initiative [34] and found similar results (Appendix A). Additionally, after the inclusion of metastasis at diagnosis as a covariate, the multivariate model showed that low FAS expression was still significantly associated with poorer overall survival, regardless of initial metastasis (Figure 1D). While low FAS expression was not significantly associated with worse event-free survival at the 0.05 level, there was still a very strong trend towards significance (Wald test *p*-value = 0.06) (Figure 1E).

### 2.2. FAS Expression Correlates with Promoter Methylation

To determine if FAS promoter methylation plays a role in regulating downstream gene expression, we examined the methylation of HM450 probes located within the FAS promoter CGI region. Using the human UCSC (GRCh37/hg19) reference genome, we identified a total of 24 FAS promoter CGI-associated probes (Figure 2A). Seven of these were classified as cross-reactive probes targeting highly repetitive sequences or homologous regions [35]. As such, they may produce an inaccurate signal and were noted. All of these CpG-island-associated probes are located in either the shelves, shores, or body of the FAS promoter CGI (chr10: 90,750,293-90,751,108). While CpG sites in the CGI body are mostly unmethylated among tumors, three probes (cg26478401, cg22936253, cg13456138) located in the CGI shores have a much higher variation (SD > 0.14) (Appendix A). We found that all three highly variant probes by methylation beta value were also strongly correlated with FAS gene expression in a negative manner (Figure 2B–D). The three probes are also significantly associated with each other based on a Spearman correlation (rho > 0.5 for all probes).

### 2.3. FAS Expression and Promoter Methylation Correlates with Metastatic Potential in Osteosarcoma Cell Lines

Next, we evaluated DNA methylation and gene expression of FAS in two osteosarcoma cell lines, SaOS-2 and LM7. SaOS-2 is the parental cell line with low metastatic potential from which the highly metastatic LM7 cell line was derived. Our data showed that SaOS-2 has a twofold higher FAS expression than LM7 (Table 1). Correspondingly, two out of the three CpG sites that displayed negative correlation with expression in tumors (cg26478401, cg13456138) also showed much lower methylation levels in SaOS-2 versus LM7 (Table 1). While we also observed negative correlation between methylation of the other CpG site (cg22936253) and FAS expression, we found that the range of methylation values for that site was small and skewed towards higher levels for all tumors and cell lines. Using bisulfite sequencing validation, we were able to confirm that the two sites, both with high beta values in LM7 and low beta values in SaOS-2, were methylated and unmethylated, respectively (Appendix A). The remaining site with high beta values in both cell lines was also methylated in both cell lines in the sequencing data. In addition to HM450 sites, we identified three additional CpG sites within 200 bp upstream of Site 1 (cg26478401) and one site downstream of Site 2 (cg22936253) that were all unmethylated in LM7 and methylated in SaOS-2 (Appendix A). In summary, the negative correlation between FAS methylation of CGI shores and expression observed among tumor samples suggests that methylation of this region may be responsible for regulating FAS gene expression.

### 2.4. Functional Studies of the Effect of FAS Demethylation

To determine the phenotypic changes as a result of alterations in FAS methylation, we used both in vitro and in vivo assays to evaluate the metastatic potential of LM7 cells following treatment with 5-aza, a demethylating agent. To confirm demethylation of FAS, LM7 cells were treated (TR) every other day for 12 days with 5-aza (5 µm) and methylation was assessed using bisulfite sequencing. Among the three HM450 sites negatively correlated with expression, Site 1 was demethylated while the other two sites remained methylated (Figure 3A, Appendix A). Notably, the three CpG sites upstream of Site 1 were all demethylated as well. These TR-LM7 cells also showed higher protein expression of FAS in comparison with untreated (UT) LM7 cells (Figure 3B). The cell migration effects of demethylating LM7 cells were tested using a wound healing assay (Figure 3C, Appendix A). LM7 cells treated with 5-aza (5 µM and 10 µM for 12 days) exhibited much slower migration in comparison to UT-LM7 cells. Whereas UT-LM7 cells had regained confluence in the scratched area after 8 days, the wounded area was still apparent in plates with TR-LM7 cells after the same time. Thus, treatment with 5-aza resulted in demethylation of several CpG sites in the north shore of the FAS promoter island and corresponded with increased protein expression and impaired cell migration.

### 2.5. Azacytidine Treatment Decreases Lung Metastasis in Orthotopic Xenograft Models

Mice injected with metastatic LM7 cells altered to re-express FAS have shown reduced development of metastasis [33]. To study the effect of FAS promoter demethylation as the mechanism of changing that expression in an in vivo model, both UT and TR-LM7 cells were injected into Nod-SCID mice orthotopically (OT). UT-LM7 cells were used as a positive control for metastasis. Survival was measured starting from the date of UT/TR-LM7 injection. Lung tissue was harvested along with bone tissue if the tumor was visibly observed in OT-injected mice and lung metastasis was confirmed histologically by the pathologist (J.H.) (Figure 4A). As expected, mice with bone tumor development (*p*-value < 0.01) and those that went on to develop lung metastasis (*p*-value = 0.0193) had significantly shorter survival compared to mice that did not develop these (Appendix A). We observed that 21% (9/42) of cases confirmed development of lung metastasis in OT-injected mice. Within these metastatic cases, only one was injected with TR-LM7 cells (Figure 4B). Using a Wald test, we found that mice injected with TR-LM7 cells had a significant improvement in survival (*p*-value = 0.036) in comparison to mice injected with UT-LM7 (Figure 4C). Based on these results, we have shown preliminary evidence that demethylation of the FAS promoter by 5-aza treatment in LM7 osteosarcoma cells may lead to an improvement in prognosis due to decreased presence of lung metastasis.

## 3. Discussion

Little improvement in osteosarcoma prognosis has been seen since the current treatment was accepted as the standard treatment regimen a few decades ago and clinical studies for novel agents has led, mostly, to mixed results [36]. The prognosis of patients with metastatic disease at diagnosis or poor responders to standard neoadjuvant treatment has remained stagnant, largely due to the lack of robust biomarkers to distinguish them from others. Using one of the largest osteosarcoma cohorts to date, we showed that FAS expression may be used as a prognostic marker for outcome in the primary tumor. We also observed that demethylation of the FAS promoter led to an increase in expression, which suggests the possibility of using demethylating agents such as decitabine to increase cell surface FAS expression in cancer cells.

Interaction between FAS and its ligand triggers FAS-induced cell death, which is one pathway through which OS cells may undergo apoptosis. Low cell surface expression of FAS may allow cancer cells to more easily bypass the checkpoint in the lungs where FASL is highly expressed. Therapeutic targets that re-express FAS in tumor cells may promote increased interaction with FASL and reduction in development of lung metastasis in OS. A clinical trial studying the addition of the macrophage and monocyte activator muramyl tripeptide (MTP) phosphatidylethanolamine (MTP-PE) and ifosfamide into standard osteosarcoma treatment identified a 7% increase in 3-year event-free survival in osteosarcoma patients [37]. Interpretation of the results from this study was complicated by an observed interaction between MTP-PE and ifosfamide. Despite this, one possible explanation for the improved prognosis in MTP-PE/ifosfamide treated patients may be due to the stimulation of regulatory cytokines such as IL-12, which upregulates FAS expression [31]. The present study demonstrates that targeting promoter methylation is another potential mechanism for regulating FAS expression. We have shown that DNA methylation in several sites located within the CpG island shore region negatively correlate with FAS expression. Demethylation of these sites in LM7, a metastatic cell line with low FAS expression, through treatment with 5-aza led to increased protein expression and decreased cell migration.

Methylation in the CGI body has long been known to affect gene expression, however, only in recent years has the association between shore or gene body methylation and cancer-related changes been examined more closely [15,16,38,39]. Previous studies have shown that FAS promoter methylation in OS was invariant and not associated with expression; however, the region studied was limited to CpG sites located in the CGI body [33]. We took advantage of the expanded coverage in the HM450 array to evaluate a more comprehensive profile of the FAS promoter CGI. We observed that all CpG sites in FAS that had high variance were located in CpG island shores and correlated negatively with FAS expression. In cutaneous T-cell lymphoma, profiling of the same region also only identified variant methylation in sites within CGI shores, including one of the same sites from the HM450 array [26]. Overall, our study provides evidence that CpG shore methylation may play a significant role in altering FAS gene expression.

Epigenetic modifications such as DNA methylation can affect the expression of key driver genes and are ideal therapeutic targets due to their reversible nature. Our study indicates that treatment of LM7 osteosarcoma cells with 5-aza resulted in demethylation of the promoter CpG sites and increased FAS protein expression compared with untreated LM7 cells. While treatment with 5-aza is expected to target genes in a non-specific manner, previous studies have demonstrated that treatment with 5-aza and decitabine affects only specific genes and more commonly affects genes in a tumor-specific environment [40,41]. Our in vivo studies demonstrated that mice injected with 5-aza treated LM7 cells formed less lung metastasis in vivo. By rescuing FAS expression through altering DNA methylation, OS cells at risk for entering the lungs and developing metastasis due to low cell surface FAS expression could now interact more often with FASL and trigger downstream apoptosis. More studies must be done to verify if the effects of FAS-specific demethylation mirror these effects. As multiple mechanisms may impact FAS expression, it is also important to consider the effect of promoter methylation in the context of other regulators such as mir-20a [42].

In summary, these results collectively demonstrate that DNA methylation in CpG island shores regulates FAS expression and restoring cell surface expression may decrease metastatic potential. FAS is likely not the only gene with epigenetic dysregulation in osteosarcoma, and further work must be done to comprehensively evaluate the methylome of osteosarcoma patients and integrate it with other genomic platforms for a more comprehensive understanding of osteosarcoma tumor biology.

## 4. Materials and Methods

### 4.1. Samples for Microarray Analyses

Matched gene expression and methylation microarray data were obtained from tumor samples collected as part of the National Cancer Institute’s Therapeutically Applicable Research to Generate Effective Treatments (TARGET) project. These samples were collected from multiple institutions including the Children’s Oncology Group (COG) and the Hospital for Sick Children in Toronto. Normal bone tissue from 2 females and 3 males obtained from Biochain (Biochain Institute Inc, Newark, CA, USA) and were used as normal controls. NHOst, a normal human osteoblast culture obtained from Clonetics (Walkerville, MD, USA), was also included in the analysis. Cell lines profiled in the microarray analysis were obtained from American Type Culture Collection (Manassas, VA, USA) and their identities have been verified by genotyping.

### 4.2. Cell Lines Used for Functional Studies

LM7 and SaOS-2 cell lines were obtained from ATCC and cultured in DMEM complete media with 10% fetal calf serum at 37 °C. LM7 cells were metastatic clones derived from the non-metastatic parental cell line SaOS-2 [43].

### 4.3. Infinium HumanMethylation450K BeadChip Assay

DNA methylation profiling was performed using Illumina’s Infinium HumanMethylation450 (HM450) BeadChip (Illumina, San Diego, CA, USA) on 86 osteosarcoma samples, 7 cell lines, and 6 normal bone tissue samples. The HM450 array profiles over 480,000 CpG sites and covers > 99% of annotated RefSeq genes and 96% of CpG islands across the genome. A 0.5 µg sample of DNA was bisulfite converted using the EZ DNA Methylation Kit (Zymo Research, Irvine, CA, USA) according to the manufacturer’s protocol. Bisulfite-treated DNA was then whole genome amplified, enzymatically fragmented, and hybridized to the BeadChip. The HM450 array assesses methylation through a single base extension step involving nucleotides fluorescently tagged with Cy3 (C,G) and Cy5 (A,T). Probe intensity after hybridization was scanned using Illumina’s iScan technology. All sample DNA was extracted at Texas Children’s Cancer Center (Houston, TX, USA) and sent to Johns Hopkins University (Baltimore, MD, USA) for processing in three batches for a total of 10 BeadChips.

All preprocessing was performed using the R (version 3.1.3) statistical programming language [44]. Quality control was performed by visual inspection of the >600 control probes for each sample on the array. No outliers were identified among control probes and no samples were excluded. Background correction was performed using the out-of-band method, which utilizes mismatched probes on the array as an estimate for background intensity [45]. Smooth quantile normalization from R’s lumi package was used to correct for color-bias between red and green channels [46]. Following this, a methylation level or beta value was calculated equal to the ratio between the methylated intensity and the overall (unmethylated plus methylated) intensity and ranges between 0 (completely unmethylated) and 1 (fully methylated). Beta-mixture quantile dilation (BMIQ) was used to normalize the distribution between Infinium I and II probes [47]. ComBat, an empirical Bayes method of adjusting for batch, was used to additively correct for technical artifacts involving scan date, row, and column location on the BeadChip array [48]. Probes with a detection *p*-value greater than 0.01, located in the X or Y chromosomes and overlapping/within 10 bp of single nucleotide polymorphisms (SNPs), were excluded from further analysis. Probes which cross-hybridized to other regions of the genome were also noted [35]. After filtering, a final set of 385,291 probes was used for downstream analysis.

### 4.4. Affymetrix Human Exon 1.0 ST Array

Sample mRNA from 89 osteosarcoma samples was analyzed using the Affymetrix GeneChip Human Exon 1.0 ST. Out of these 89 cases, 86 had matched methylation data. Robust multi-array average (RMA) normalization was performed in Affymetrix Power Tools (APT, v1.17.0) to generate normalized gene summarized expression values based on the hg19 reference genome [49]. RMA was used to perform background adjustment, quantile normalization, summarization of probes into probe sets, and calculate log2 transformed expression values. Sample quality control (QC) metrics were analyzed using APT. No samples were excluded after QC.

### 4.5. Bisulfite-Specific Sequencing

DNA (200–500 ng) from LM7 and SaOS-2 osteosarcoma cell lines was extracted and bisulfite converted using the EZ DNA Methylation Kit, according to the manufacturer’s instructions. Primers for two regions overlapping the three CpG sites of interest identified on the HM450 BeadChip are as follows:Region 1 Forward, 5′-GTTTTAAAGTAATAGTGATTTTGAATAGTG-3′Region 1 Reverse, 5′-CTAAAAAATTAAAAAAATCTTAAAAAAAA-3′Region 2 Forward, 5′-AATGTAGATGAGTTAAATATAAA GATTAGA-3′Region 2 Reverse, 5′-AAATCCATAAACTCTTAAAAACTTC-3′

For PCR amplification, bisulfite-modified DNA was combined with 50 µL of a mixture containing 10× PCR buffer, 2.0 mM MgCl_2_, 10 mM dNTP, 1 µL upstream, 1 µL downstream primer (at 1 µg/µL stock), and 800 U/µL FastStartTaq DNA polymerase (Roche Applied Science, Mannheim, Germany). PCR conditions were optimized from the manufacturer’s guidelines (Applied Biosystems GeneAmp PCR System 9600, Waltham, MA, USA): 1 cycle of 95 °C for 5 min, 40 cycles of 95 °C for 30 s, 55 °C for 40 s, 15 cycles of 72 °C for 1 min to 3 min, and a final extension step for 7 min at 72 °C. Following this, PCR products were subcloned into pcDNA3.1 and 3 clones were chosen for downstream sequencing. Treated LM7 cells were incubated with 5 µM 5-aza every other day for 11 days and sequenced with the same method above.

### 4.6. Wound Healing Assay of 5-aza Treated LM7 Cells

Three sets of 5-aza-treated LM7 cells (described previously) were seeded in triplicate in 96-well plates and grown to confluence. Two sets of LM7 cells were treated with 5-aza at 5 µM and 10 µM following the same treatment regimen described above. The well was scratched with a sterile micropipette tip to create a zone of constant width. Cells were allowed to re-infiltrate the cleared region over the course of 8 days and were observed using phase contrast microscopy and photographed at Day 0, Day 1, Day 4, and Day 8 to track cell migration. Untreated LM7 cells were used as control.

### 4.7. Orthotopic Xenograft Mouse Study

The 5-week-old nod/SCID/γ mice were obtained from Jackson Labs (The Jackson Laboratory, Bar Harbor, ME, USA). Mice were orthotopically injected in the proximal tibia with a total number of 5 × 10^5^ cells that had been treated with (n = 15) or without (n = 27) 10 µM 5-aza every other day for 11 days. Mice were sacrificed at sign of sickness and lung tissue and/or bone tumor tissue were collected, formalin fixed, and paraffin embedded for histopathologic analysis.

### 4.8. Statistical Analysis

All statistical analyses were performed in R using the stats and survival packages. A Student’s t-test was used to compare expression and methylation values between clinical covariates. Spearman correlation was used to examine the association between expression log2 intensity values and methylation beta values. Cox proportional hazards regression models were used for survival analysis. Kaplan–Meier estimates were used to generate survival curves. A *p*-value of 0.05 was considered as passing significance for all analysis. Correction for multiple testing was performed using the Benjamini–Hochberg (BH) false-discovery rate correction method [50].

## Figures and Tables

**Figure 1 ijms-24-12155-f001:**
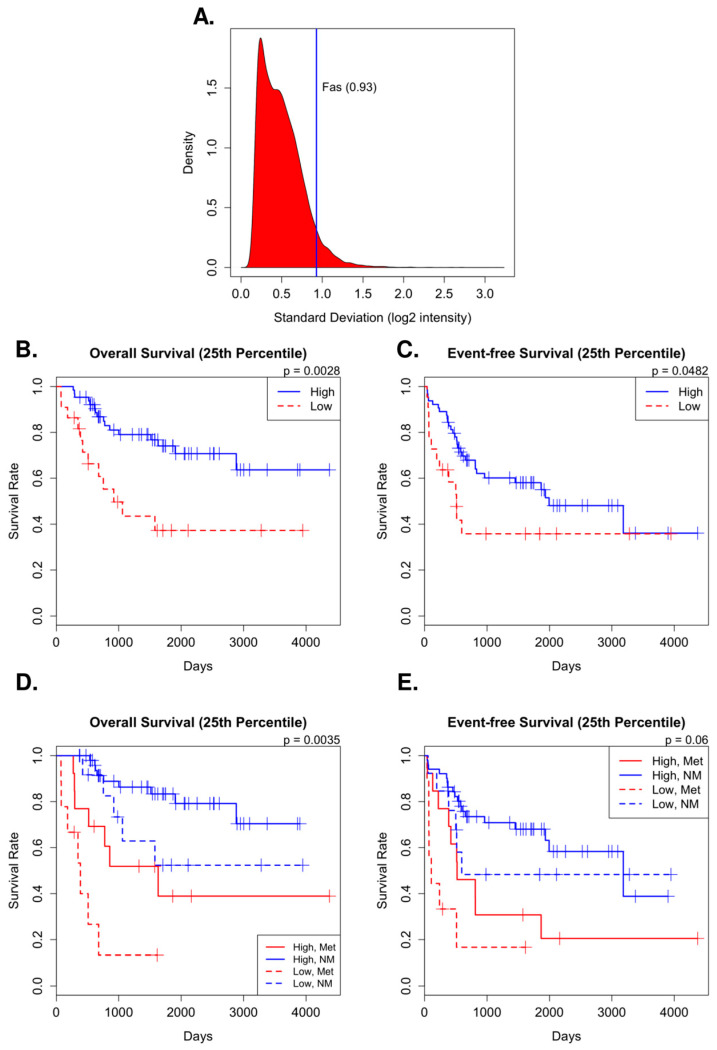
Variant FAS expression in osteosarcoma tumor samples is associated with outcome. (**A**) Comparison between standard deviation among tumor samples of mRNA expression (log2 intensity) of FAS versus all other genes on the expression array. (**B**–**E**) Kaplan–Meier survival curves comparing high and low FAS mRNA (stratified at the 25th percentile). Overall survival and event-free survival are calculated using (**A**,**B**), only FAS mRNA high or low expression status and (**D**,**E**), after including metastasis at diagnosis status as a covariate. Significant differences in survival calculated using a Wald test. Only the *p*-value of FAS expression is reported in the case of multivariate models.

**Figure 2 ijms-24-12155-f002:**
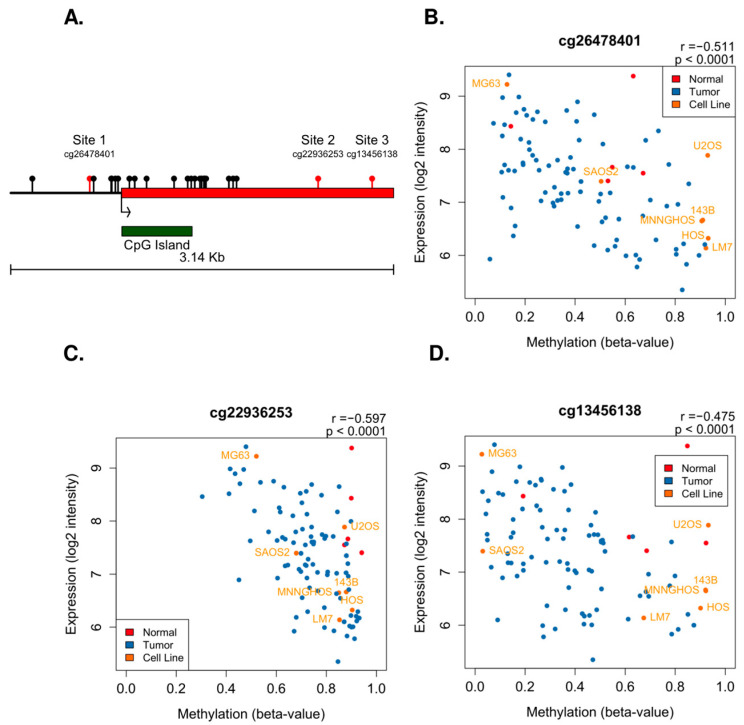
FAS promoter methylation negatively correlates with mRNA expression. (**A**) Relative location of all HM450 probes overlapping FAS promoter and start of coding region. Red indicates methylation probes significantly correlated with FAS mRNA expression in a negative manner. The red bar indicates the coding region. (**B**–**D**) Plots showing correlation between methylation beta value and expression log2 intensity for three sites with significant negative correlation.

**Figure 3 ijms-24-12155-f003:**
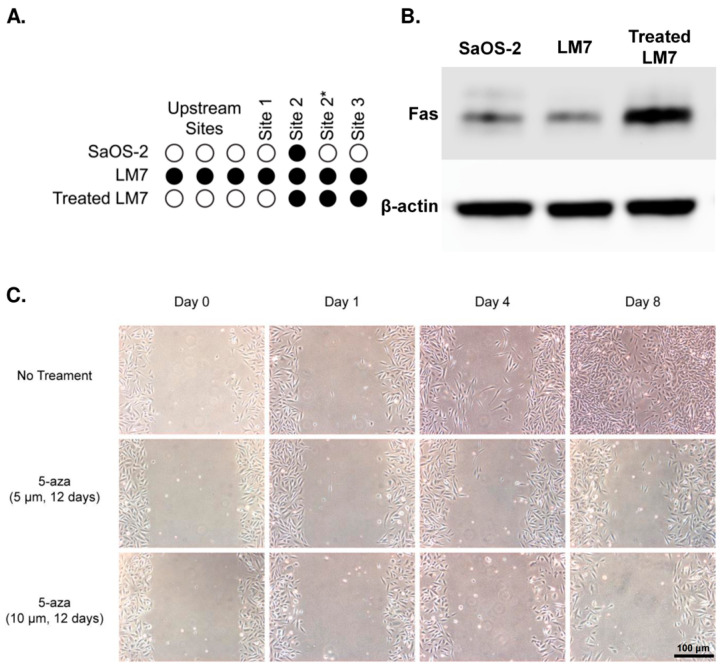
Functional effects of FAS demethylation. (**A**) Bisulfite sequencing validation results from SaOS-2 and LM7 cell lines. Site 2* represents a CpG site 6bp upstream of Site 2. (**B**) Western blot of FAS protein expression shows TR-LM7 cells have higher protein expression in comparison with UT-LM7 cells. (**C**) TR-LM7 cells at 5 µM and 10 µM for 12 days were compared with UT-LM7 cells in a wound healing assay.

**Figure 4 ijms-24-12155-f004:**
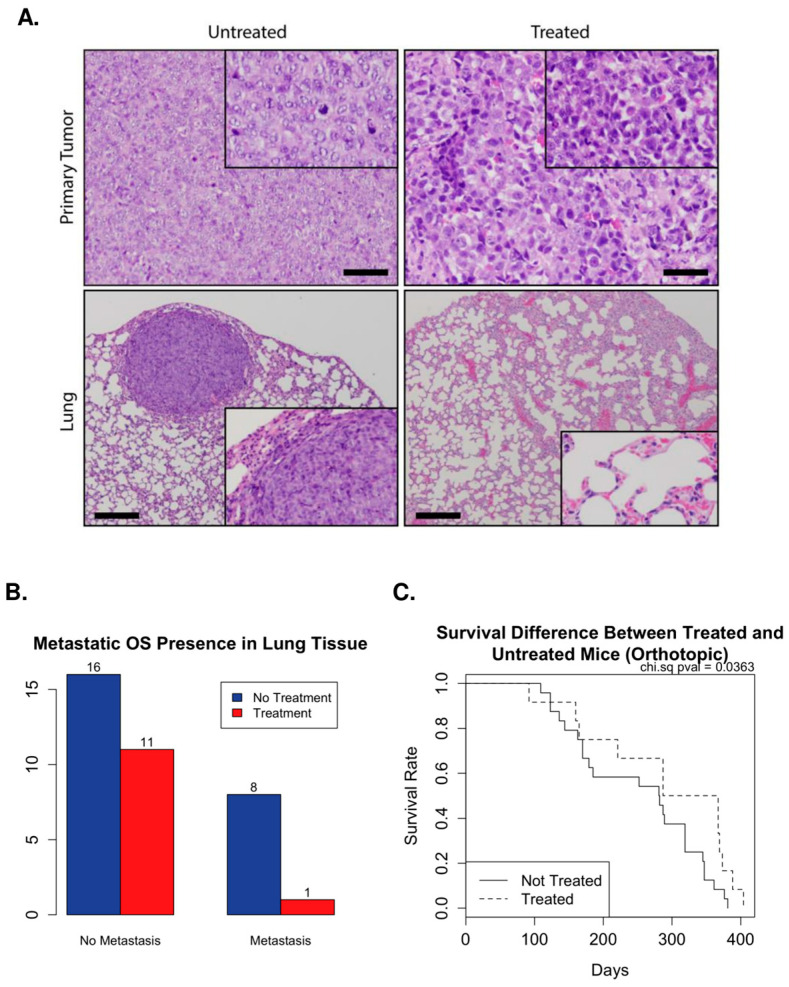
Orthotopic injection of 5-aza TR and UT-LM7 cells. (**A**) Primary bone tumor and lung tissues with and without metastasis from OT-injected mice. Mice injected with UT-LM7 cells developed both primary tumor and lung metastasis while injection with TR-LM7 cells resulted in fewer metastases, even after development of bone tumor. Scale bar in zoomed out image = 50 µm, zoomed in image = 25 µm. (**B**) Mice injected with TR-LM7 cells trend towards a low number of pulmonary metastases. (**C**) Mice injected with TR-LM7 have a significantly improved overall survival rate compared to those injected with UT-LM7 based on a Wald test (*p*-value = 0.036).

**Table 1 ijms-24-12155-t001:** Summary of FAS expression and methylation values in osteosarcoma cell lines.

OS Cell Line	Expression ^a^	Methylation		
	‘3257098’	cg26478401	cg22936253	cg13456138
LM7	6.138	0.922	0.853	0.674
SaOS-2	7.395	0.504	0.680	0.030

^a^ mRNA expression of FAS analyzed using probe set ‘3257098’ as indicated by Affymetrix.

## Data Availability

Methylation and gene expression array data can be found at the National Cancer Institute’s Genomic Data Commons (https://portal.gdc.cancer.gov/projects/TARGET-OS, accessed on 27 July 2023).

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
