# Peer review of "The Role of FAS Receptor Methylation in Osteosarcoma Metastasis"

_ijms, 2023, doi:10.3390/ijms241512155_

Round 1

Reviewer 1 Report

This is a good paper describing FAS methylation and methylation regulation.

It is an excellent experimental study with new data and results. 

Some comments

1. The discussion does not have much information on how common alterations of FAS and FASL expression in osteosarcoma or other sarcomas and carcinomas are.

2. Osteoasraciom has a couple of studies describing the gene expression profiles and integrative genomic data analysis (PMID: 29050494, 25496518). Maybe authors can check these articles to get comparative data for their manuscript.

3. This comparison is important to improve the translational potential of the mansucript.

Reviewer 2 Report

Dear Authors,

The role of FAS receptor methylation in osteosarcoma metastasis by Sun et al is interesting. However, the current version of the manuscript needs some additional data or information to improve the quality of the manuscript to publish in IJMS journal.

Major comments.

Comments: 1.  Authors could show the % of wounder closer (graphical representation) compared to control, 5 mM and 10 5 mM in Figure 3 will be nice.

Comments: 2.    Nice to see the 5-Aza kill curve at least in cells.

Comments: 3.   Please include vehicle injected mice (control with no tumor mice) in IHC. This will help to see the real survival of the mice.

Comments: 4.  The authors using 5-Aza to inhibit DNMT activity. It will be nice to see the DNMT1 and DNMT3 immunoblot before after the treatment in cells as well as in tumor samples.

Minor English correction needed

Round 2

Reviewer 2 Report

The manuscript is improved

Author Response

Dear reviewer,

Thank you for your positive review.

Sincerely,
Aaron Taylor